# ULF: Unsupervised Labeling Function Correction using Cross-Validation for Weak Supervision

## Abstract

A way to overcome expensive and time-consuming manual data labeling is *weak supervision* - automatic annotation of data samples via a predefined set of labeling functions (LFs), rule-based mechanisms that generate artificial labels for the classes associated with the LFs. In this work, we investigate noise reduction techniques for weak supervision based on the principle of $k$-fold cross-validation. We introduce a new algorithm *ULF* for denoising weakly annotated data which uses models trained on all but some LFs to detect and correct biases specific to the held-out LFs. Specifically, ULF refines the allocation of LFs to classes by re-estimating this assignment on highly reliable cross-validated samples. We realize two variants of this algorithm: feature-based ULF (relying on count-based feature vectors), and DeepULF (fine-tuning pre-trained language models). We compare ULF to methods originally developed for detecting erroneous samples in manually annotated data, as well as to our extensions of such methods to the weakly supervised setting. Our new weak supervision-specific methods (ULF and extensions) leverage the information about matching LFs, making detecting noisy samples more accurate. Evaluation on several datasets shows that ULF can successfully improve weakly supervised learning without utilizing any manually labeled data.

## 1 Introduction

A large part of today's machine learning success rests upon a vast amount of annotated training data. However, a manual expert annotation turns out to be tedious and expensive work. There are different approaches to reduce this data bottleneck: fine-tuning large pre-trained models (Devlin et al., 2019), applying active learning (Sun & Grishman, 2012) and semi-supervised learning (Kozareva et al., 2008). However, even if in a reduced amount, these approaches still demand manually annotated data. Moreover, constant data re-annotation would be necessary in settings with dynamically changing task specifications or changing data distributions.

Another strategy that does not require any manual data labeling is *weak supervision* (WS), which allows to obtain massive amounts of labeled training data at a low cost. In a weakly supervised setting, the data is labeled in an automated process using one or multiple weak supervision sources, such as external knowledge bases (Lin et al., 2016; Mintz et al., 2009) and manually-defined or automatically generated heuristics (Varma & Ré, 2018). By applying such rules, or *labeling functions* (LFs, Ratner et al., 2020), to a large unlabeled dataset, one can quickly obtain weak training labels, which are, however, potentially error-prone and need additional denoising (see examples in Fig. 1).

In this work, we explore methods for improving the quality of weak labels using methods based on the principle of *$k$-fold cross-validation*. Intuitively, if some part of the data is left out during training, the model does not overfit on errors specific to that part. Therefore, a mismatch between predictions of a model (trained on a large portion of the data set) and labels (of the held-out portion) can indicate candidates of noise specific to the held-out portion.

This idea has motivated different approaches to data cleaning (Northcutt et al., 2021; Wang et al., 2019c). As they deal with general, non-weakly supervised data, they usually split the data samples into folds *randomly* and *independently* on the sample level. However, a direct application of these methods to weakly labeled data ignores valuable knowledge stemming from the weak supervision

process (e.g., which LFs matched in each sample or what class each LF corresponds to). In this work, we leverage this additional source of knowledge by splitting the data considering the LFs matched in the samples. We build on the intuition that a mismatch between predictions of a model (trained with a large portion of the LFs) and labels (generated by held-out LFs) can indicate candidates of noise specific to the held-out LFs. If this LF-specific cross-validation is done for *each* LF, noise associated with all LFs could be found and corrected. This idea is realized in our extensions to methods proposed in Northcutt et al. 2021 and Wang et al. 2019c.

Apart from that, we use the principle of weakly supervised cross-validation to break out of the logic of just repairing the labels, to instead repair the LF-to-class assignment. This approach is formalized in **ULF** - our new method for **U**nsupervised **L**abeling **F**unction correction with $k$-fold cross-validation. Its primary goal is to improve the *LFs to classes allocation* in order to correct the systematically biased label assignments. ULF re-estimates the joint distribution between LFs and class labels during cross-validation based on highly confident class predictions and their co-occurrence with matching LFs. The improved allocation allows to re-assign the weak labels for further training. Importantly, ULF also improves labels of the samples with no LFs matched, in contrast to other methods that filter them out (Ratner et al., 2020).

| | Matched labeling functions | Assigned Label |
|---|---|---|
| (1) if your like drones, **plz subscribe** to Kamal Tayara. He takes videos with his drone that are absolutely beautiful. | keyword("subscribe") keyword("plz") | SPAM |
| (2) It looks so real and **my** daughter is a big fan and she likes a lot of your **songs.** | keyword("my") keyword("song") | SPAM/ HAM |
| (3) Follow me on Twitter @mscalifornia95 | no matches | — |

Figure 1: Examples of weakly supervised annotation from YouTube dataset. In (1), both matched LFs correspond to the SPAM class; the sample is therefore assigned to SPAM class. In (2), there is a conflict as one of the matched LFs belongs to the SPAM class, while the other - to the HAM class. In (3), no LFs matched, meaning the sample does not get any weak signal.

Overall, our main contributions are: **(1)** A new method **ULF** for improving the LFs to classes allocation in an unsupervised fashion. ULF not only detects inconsistent predictions but corrects the process that led to them by re-estimating the assignment of LFs to classes. Training with ULF results in more accurate labels and a better quality of the trained classifier. **(2)** Two implementations of ULF: feature-based ULF for feature-based learning (without a hidden layer), and DeepULF for fine-tuning pre-trained language models. **(3)** Extensions of two methods for denoising the data using the principle of $k$-fold cross-validation (Wang et al., 2019c; Northcutt et al., 2021). Our extensions Weakly Supervised CrossWeigh and Weakly Supervised Cleanlab profit from the WS-specific information and make the denoising of WS data more accurate. **(4)** Extensive experiments on several weakly supervised datasets in order to demonstrate the effectiveness of our methods. To the best of our knowledge, we are the first (1) to adapt $k$-fold cross-validation-based noise detection methods to WS problems, and (2) to refine the LFs to classes allocation in the WS setting.

## 2 RELATED WORK

Weak supervision has been widely applied to different tasks in various domains, such as text classification (Ren et al., 2020; Shu et al., 2020), relation extraction (Yuan et al., 2019; Hoffmann et al., 2011), named entity recognition (Lan et al., 2020; Wang et al., 2019c), video analysis (Fang et al., 2020; Kundu et al., 2019), medical domain (Fries et al., 2021), image classification (Li et al., 2021), and others. Weak labels are usually cheap and easy to obtain, but also potentially error-prone, and thus often need additional denoising.

**Denoising methods.** Among the most popular approaches to improving weakly supervised data is building a specific model architecture or reformulating the loss functions (Karamanolakis et al., 2021; Hedderich & Klakow, 2018; Goldberger & Ben-Reuven, 2017; Sukhbaatar et al., 2014). Sometimes, weak labels are combined with additional expert annotations: for example, by adding

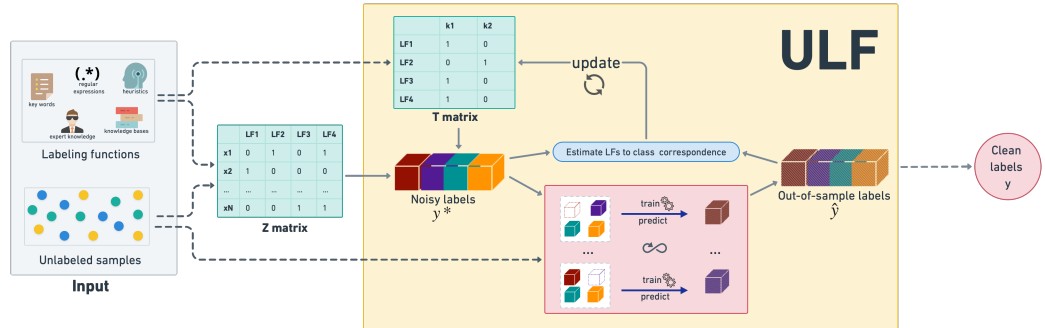

Figure 2: ULF: Unsupervised Labeling Function Correction

a set of manually annotated data to the weakly labeled one (Mazzetto et al., 2021; Karamanolakis et al., 2021; Awasthi et al., 2020), or learning from manual user corrections or guidance (Hedderich et al., 2021; Chatterjee et al., 2020; Boecking et al., 2021; Lange et al., 2019). The methods we introduce in this paper, on the contrary, do not require any manual supervision and can be used with any classifier. Other approaches aim at finding and removing or downweighting erroneous samples by estimating the relation between noisy and clean labels (Wang et al., 2019a; Northcutt et al., 2021). Rather than discarding potentially erroneous samples, our method ULF aims at *label correction*, utilizing as much of the weakly supervised data as possible.

**Cross-Validation.** There are several methods that share the idea of using $k$-fold cross-validation for denoising manually labeled data (Wang et al., 2019b;c; Northcutt et al., 2021; Teljstedt et al., 2015). Being a stable and reliable method of trained model quality estimation (Wong & Yeh, 2019), $k$-fold cross-validation has proven to be a fruitful method for detecting errors in data as well. Wang et al. 2019c denoise crowdsourcing annotations by repeatedly training $k$ models and downweighting the samples with disagreeing out-of-sample predicted and original labels. Northcutt et al. 2021 consider the confidence of the predicted class probabilities, identify the mislabeled samples using ranking and prune them. However, being used *as-is* for denoising the weakly supervised data, both of these methods miss a lot of potentially profitable WS-specific information.

**Data Programming.** We follow the data programming approach (Ratner et al., 2016) which allows encoding the weak supervision sources in form of labeling functions (LFs) which could be viewed as annotators *annotating* data samples. Such annotations often result in conflicting or erroneous labels, which are usually refined by applying more sophisticated LFs aggregation techniques (Ratner et al., 2019; Hoffmann et al., 2011) and modifying the LFs (Chatterjee et al., 2019).

## 3  ULF: Unsupervised Labeling Function Correction

Inaccurate and erroneous allocation of LFs to classes often becomes a cause of noise in a weakly supervised setting. For example, in Figure 1, among the LFs used to annotate the YouTube Dataset, there is a LF *"my"* that is set to correspond to the SPAM class. The dataset creators (Alberto et al., 2015) included this LF to capture the often encountered spam messages like *"subscribe to **my** channel"* or *"follow **my** page"*. However, this correspondence is not that straightforward and may be potentially misguiding; it is not hard to come up with a dozen non-spam messages containing the word *"my"*. Thus, the weak labeling for Sample 2, where this LF matched alongside another one from class HAM (i.e., a non-SPAM class), would lead to a tie and, by a random tie-breaking, potentially to an incorrect label assignment. However, if the class assignment of *"my"* were adjusted so that its association with the SPAM class is reduced, the overall HAM probability would dominate, and a correct label would be assigned to this sample.

### 3.1  PRELIMINARIES

Given a dataset $X$ with $N$ data samples, $X = \{x_1, x_2, ..., x_N\}$; each sample is a sequence of words (i.e., one or several sentences). This set is used for training a classifier with $K$ output classes.

In the weakly supervised setting, there are no *known-to-be-correct* labels for the training samples. Instead, we are provided with a set of *LFs* $L$, $L = \{l_1, l_2, ..., l_L\}$. The LFs can express different weak supervision sources, such as knowledge bases, lists of keywords and other human-defined heuristics. We say that a LF $l_j$ *matches* a sample $x_i$ if some condition formulated in $l_j$ holds for $x_i$ (e.g. a keyword or a regular expression matches); then $l_j$ maps $x_j$ to a label. For example, in the case of keyword-based LFs, a LF matches a sample if the keyword is present in it and, thus, a corresponding label is assigned to this data sample. A set of LFs matched in a sample $x_i$ is denoted by $L_{x_i}$, where $|L_{x_i}| \in [0, L]$. This information can be saved for an entire data set in a binary matrix $Z_{N \times L}$, where $Z_{ij} = 1$ means that $l_j$ matches in sample $x_i$.

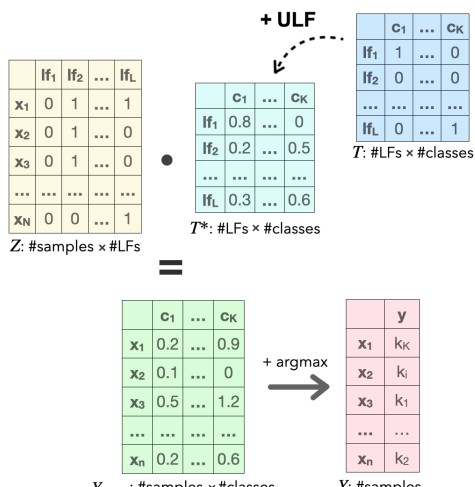

Figure 3: Weak annotation encoded with Z and T matrices. $Z$ contains the information about LFs matches in samples. $T$ is a matrix with LFs-to-class correspondence. Multiplication of $Z$ and $T$ matrices and applying majority vote results in weak labels $\tilde{Y}$. $T^*$ is the $T$ matrix improved with ULF.

Each LF $l_j$ corresponds to some class $k_i$; in other words, $l_j$ *assigns* the corresponding label to samples where $l_j$ matches[1]. The information about this correspondence is stored in a binary matrix $T_{L \times K}$, where $T_{ij} = 1$ means LF $l_i$ corresponds to class $j$. The most straightforward way of obtaining weak labels $\tilde{Y} = \{\tilde{y}_1, \tilde{y}_2, ..., \tilde{y}_n\}$, $\tilde{y}_j \in K$ is to multiply $Z$ and $T$, apply majority vote, and break the ties.

## 3.2 OVERVIEW

The main goal of the ULF algorithm is to refine the matrix of LFs to classes assignments $T$. The main steps are the following: first, the predictions (together with their confidences) are calculated with $k$-fold cross-validation. Second, these predictions are used to build a LFs-to-classes confident matrix $C_{L \times K}$. This matrix counts how many samples with a particular LF have highly confident predictions (exceeding a threshold) for a particular class. $C$ is then re-normalized to provide a cross-validated re-estimation of $T$. The graphical illustration of the algorithm is provided in Figure 3, the method is summarized in Algorithm 1.

## 3.3 LABEL PROBABILITIES WITH CROSS-VALIDATION

Firstly, class probabilities for each training sample are predicted by *k*-fold cross-validation. For that, we use the training set $X$ and weak labels $\tilde{Y}$ obtained by multiplying $Z$ and $T$ matrices. There are three possible ways of splitting the data in folds for cross-validation training:

- **randomly (ULF_{rndm})**: the samples are assigned to folds the same way as it would be done in standard *k*-fold cross-validation irrespective of LFs matching;
- **by LF (ULF_{lfs})**: the LFs are randomly split into $k$ folds $\{f_1, ..., f_k\}$ and each fold $f_i$ is iteratively taken as held-out LFs, while others become training LFs. All samples where training LFs match become training samples, the rest are used for re-estimation: $X_{train_i} = \{x_j | L_{x_j} \cap f_i = \emptyset\}$, $X_{out_i} = X \setminus X_{train_i}$  (1).
- **by signatures (ULF_{sgn})**: for each training sample $x_i$ we take the set of matching LFs $L_{x_i}$ as its signature and build $k$ folds from them, each of which iteratively be-

---

[1]The initial manual class assignment is typically such that a LF corresponds to exactly one class, for which it covers a prototypical case. A manual assignment to several classes would however be possible in principle and compatible with ULF.

comes a held-out fold, while others are used as training folds: $X_{train_i} = \{x_j | L_{x_j} \notin f_i\}$, $X_{out_i} = \{x_j | L_{x_j} \in f_i\}$ (2).

Secondly, $k$ models are separately trained on $X_{train_i}$, $i \in [1, k]$ and applied on the held-out folds $X_{out_i}$ to obtain the out-of-sample predicted probabilities $P_{N \times K}$. From $P_{N \times K}$ the out-of-sample labels $\hat{y}$ are derived:

$$\hat{y}_i = \underset{1 \leq j \leq K}{\arg \max} \, p(\tilde{y} = j; x_i, \theta) \tag{3}$$

Importantly, a $x_i$ sample probability $p(\tilde{y} = j; x_i, \theta)$ is considered **only** if it exceeds the class $j$ average threshold $t_j$:

$$t_j := \sum_{x_i \in X_{\tilde{y}=j}} p(\tilde{y} = j; x_i, \theta) / |X_{\tilde{y}=j}| \tag{4}$$

This allows handling of the sample $x_i$ as belonging to the class $j$ if and only if this sample *confidently* belongs to the corresponding class. If no probability exceeds the class thresholds for a sample (e.g., it belongs to all classes with equally small probabilities), the sample is disregarded in further calculations as unreliable.

### 3.4 RE-ESTIMATE LABELING FUNCTIONS

In order to refine the LFs to classes allocation, ULF re-estimates the joint distribution between *matched LFs* and *predicted labels*. For each LF $l_i$ and each class $k_j$, the number of samples is calculated, which have the LF $l_i$ matched and were confidently assigned to the class $k_j$. This information is saved as a *LFs-confident matrix* $C_{L \times K}$ that contains the counts of samples:

$$C_{l_i, \hat{y}_j} = |\{x_i \in X : \hat{y}_i = \tilde{y}_j, l_i \in L_{x_i}\}| \tag{5}$$

Next, the confident matrix is calibrated and normalized to $\hat{Q}_{L \times K}$ to correspond to the data proportion in the $Z$ matrix:

$$\hat{Q}_{l_i, \hat{y}_j} = \left( C_{l_i, \hat{y}_j} \cdot \sum_{m=1}^{L} Z_{l_m, \hat{y}_j} \right) / \left( \sum_{m=1}^{L} C_{l_m, \hat{y}_j} \right), \tag{6}$$

where $\sum_{\substack{i \in L, \\ j \in K}} C_{l_i, \hat{y}_j} = n$, $\sum_{j=1}^{K} Z_{l_m, \hat{y}_j} = \sum_{j=1}^{K} \hat{Q}_{l_m, \hat{y}_j}$.

The joint matrix $\hat{Q}$ can now be used for improving the LFs-to-class matrix $T$ that contains the initial LFs to class allocations. $T$ and $\hat{Q}$ are summed with multiplying coefficients $p$ and $1 - p$, where $p \in [0, 1]$. The value of $p$ balances the initial manual label assignment with the unsupervised re-estimation and determines how much information from the estimated assignment matrix $\hat{Q}$ should be preserved in the refined matrix $T^*$:

$$T^* = p * \hat{Q} + (1 - p) * T \tag{7}$$

With the multiplication of $Z$ and the newly recalculated $T^*$ matrices, an updated set of labels $Y^*$ is calculated. It can be used either for rerunning the denoising process or for training the end classifier.

### 3.5 FEATURE-BASED ULF VS DEEPULF

We propose two ways of realizing the unsupervised label correction procedure described above. The first variation of the ULF algorithm is **feature-based ULF**, which uses count-based features. The training steps (i.e., Alg. 1, lines 7 and 14) are realized with Algorithm 2, where any model for the feature-based prediction can be used. The second one - **DeepULF** - fine-tunes pre-trained language models. The fine-tuning can be combined with an additional contrastive self-learning step from the Cosine framework (Yu et al., 2021), which leverages unlabeled data and potentially improves the quality of the trained classifier. The training steps (Alg. 1, lines 7 and 14) for DeepULF are summarized in Algorithm 3. As signature-based data splitting was the best performing method in the experiments with feature-based ULF, we select it for the DeepULF experiments.

**Algorithm 1:** ULF: Unsupervised Labeling Function Correction for Weak Supervision

**Input:** unsupervised training data $X$,
samples to LFs matrix $Z_{N \times L}$,
LFs to classes matrix $T_{L \times K}$

1 Calculate noisy labels $\tilde{Y} \leftarrow Z * T$
2 **for** $iter = 1, 2, ...I$ **do**
3     Randomly split LFs $L$ or signatures $\bigcup L_{x_i}$ into $k$ folds $f_1, ..., f_k$
4     **for** $f = 1, 2, ..., k$ **do**
5        Build $X_{train_i}, X_{out_i}$ sets (Eq. 1, 2)
6        $\tilde{Y}_{train_i} = \{\tilde{y} \in \tilde{Y} : \forall x \in X_{train_i}\}$
7        Train $(\theta_i, X_{train_i}, \tilde{Y}_{train_i})$
8        Calculate $p(\tilde{y} = j; x_i, \theta_i)$
9     Get labels $\hat{y}_i$ (Eq. 3) with respect to the thresholds $t_j$ (Eq. 4)
10     Calculate LFs-to-class confident matrix $C_{l,\hat{y}}$ (Eq. 5)
11     Estimate a joint matrix $\hat{Q}_{l,\hat{y}}$ (Eq. 6)
12     Recalculate $T^*$ matrix (Eq. 7)
13     Calculate new noisy labels $\tilde{Y}^*$ with $T^*$
14 Train $(\theta, X, \tilde{Y}^*)$

**Output:** Trained $\theta$

**Algorithm 2:** Train $(\theta, X, \tilde{Y})$: Feature-based ULF

1 $\theta = \text{AdamW}(\theta, X, \tilde{Y})$

**Output:** Trained $\theta$

**Algorithm 3:** Train $(\theta, X, \tilde{Y})$: DeepULF; pretrained language model fine-tuning (optionally: with Cosine self-training)

1 $X_{lab} = \{x \in X : |L_{x_i}| > 0\}$
2 $\tilde{Y}_{lab} = \{\tilde{y} \in \tilde{Y} : \forall x \in X_{lab}\}$
    # 1. fine-tune $\theta$ with $X_{lab}$ and $\tilde{Y}_{lab}$
3 $\theta = \text{AdamW}(\theta, X_{lab}, \tilde{Y}_{lab})$
    # 2. (optional) contrastive self-training of $\theta$ with $X$
4 Calculate pseudo labels $y_{\text{psd}}$
5 **for** $step = 1, 2, ..., num\_steps$ **do**
6     Select confident samples
7     Calculate classification loss $L_c(\theta, y_{\text{psd}})$, contrastive regularizer $R_1(\theta, y_{\text{psd}})$, confidence regularizer $R_1(\theta)$
8     $L(\theta, y_{\text{psd}}) = L_c + \lambda R_1 + R_1$
9     $\theta = \text{AdamW}(\theta, X)$

**Output:** Trained $\theta$

**Unlabeled instances.** One of the challenges in weak supervision is the data samples where no LF matches; in other words, the samples without any weak signal. In some approaches, such samples are filtered out, while in others they are kept as belonging to a random or the *other* class. In feature-based ULF, such samples are initially assigned with random labels and partly involved in cross-validation training. The number of samples to be used in cross-validation training is defined by hyper-parameter $\lambda$. After each $T^*$ matrix recalculation, their new labels are calculated directly from the out-of-samples predicted probabilities with respect to the class average threshold (see Eq. 3 and 4). In DeepULF combined with Cosine such samples are leveraged during self-supervised learning.

## 4 WSCrossWeigh and WSCleanLab

In order to have suitable baselines, we consider two other cross-validation-based methods initially proposed to improve manual data annotations: CrossWeigh (Wang et al., 2019c), and Cleanlab (Northcutt et al., 2021). For a more detailed comparison, we introduce extensions of these frameworks for the weakly supervised setting. In contrast to the original methods, our extensions leverage the LF information resulting in more accurate results.

CrossWeigh was proposed for tracing inconsistent labels in the crowdsourced annotations for named entity recognition. The algorithm traces the repeatedly mislabeled entities by training multiple models in a cross-validation fashion on different data subsets, comparing the predictions on unseen data with its crowdsourced labels and reweighting the conflicting ones. Our **Weakly Supervised Cross-Weigh** (WSCW) splits the data with respect to the LFs (by Eq. 1) and trains the $k$ separate models on $X_{train_i}$. The labels $\hat{y}$ predicted by the trained model for the samples in $X_{out_i}$ are then compared to the initial noisy labels $\tilde{y}$. All samples $X_j$ where $\hat{y}_j \neq \tilde{y}_j$ are claimed to be potentially mislabeled; their influence is reduced in further training. The whole procedure of errors detection is performed $t$ times with different partitions to refine the results. More details are provided in Appendix A.

The Cleanlab framework allows to find erroneous labels by estimating the joint distribution between the noisy labels and out-of-sample labels calculated by $k$-fold cross-validation. In our extension **Weakly Supervised Cleanlab** (WSCL), we follow a similar approach, but adapt it to the weak supervision the same way as in WSCW: the cross-validation sets $X_{train_i}$ and $X_{out_i}$ sets are built considering the matched LFs by Eq. 1 (WSCL $_{lfs}$) or Eq. 2 (WSCL $_{sgn}$). These sets are used to train $k$ models, which predict the class probabilities for the held-out samples, which are later transformed

| | YouTube (Acc) | Spouse (F1) | TREC (Acc) | SMS (F1) | Avg |
|---|---|---|---|---|---|
| **Manually Supervised** | | | | | |
| Gold | $94.0 \pm 0.0$ | - | $81.9 \pm 0.5$ | $91.6 \pm 0.2$ | 89.3 |
| **Weakly Supervised** | | | | | |
| Majority Vote | $86.8 \pm 0.5$ | $41.2 \pm 0.2$ | $55.9 \pm 0.1$ | $80.0 \pm 0.2$ | 66.0 |
| Snorkel-DP (Ratner et al., 2020) | $88.0 \pm 0.1$ | $41.5 \pm 0.7$ | $42.2 \pm 2.2$ | $82.8 \pm 0.1$ | 63.6 |
| **Denoising** | | | | | |
| CrossWeigh (Wang et al., 2019c) | $90.1 \pm 0.6$ | $41.6 \pm 0.5$ | $40. \pm 0.1$ | $79.6 \pm 1.4$ | 62.8 |
| Cleanlab (Northcutt et al., 2021) | $78.0 \pm 0.2$ | $\underline{45.1 \pm 0.1}$ | $58.5 \pm 0.1$ | $79.5 \pm 0.2$ | 65.3 |
| CL-PyTorch | $86.9 \pm 0.7$ | $44.4 \pm 1.2$ | $55.8 \pm 0.3$ | $86.0 \pm 0.6$ | 68.3 |
| **WS Extentions** (Ours) | | | | | |
| WSCW | $90.8 \pm 0.7$ | $42.0 \pm 0.0$ | $46.0 \pm 0.4$ | $84.0 \pm 0.9$ | 65.7 |
| WSCL $_{lfs}$ | $87.2 \pm 0.4$ | $44.4 \pm 0.7$ | $\textbf{58.9} \pm 0.3$ | $\underline{86.6 \pm 0.3}$ | 69.3 |
| WSCL $_{sgn}$ | $88.3 \pm 0.5$ | $44.6 \pm 0.9$ | $56.4 \pm 0.3$ | $85.1 \pm 0.3$ | 68.6 |
| **ULF** (Ours) | | | | | |
| ULF $_{rndm}$ | $\underline{92.8 \pm 0.1}$ | $44.8 \pm 0.5$ | $58.0 \pm 0.2$ | $85.7 \pm 0.5$ | $\underline{70.3}$ |
| ULF $_{lfs}$ | $90.8 \pm 0.9$ | $44.0 \pm 0.9$ | $55.5 \pm 0.4$ | $70.0 \pm 0.4$ | 65.1 |
| ULF $_{sgn}$ | $\textbf{94.6} \pm 0.2$ | $\textbf{49.8} \pm \textbf{1.0}$ | $58.2 \pm 0.2$ | $\textbf{88.6} \pm 0.3$ | **72.8** |

Table 1: Feature-based ULF results. The best results are marked **bold**, the second best - underlined. All results are averaged over 10 trials and reported with the standard error of the mean.

to the exact labels $\hat{y}$ with respect to the class expected value. The correspondence between noisy and out-of-sample predicted labels determines the number of samples to be pruned (the same way as in the default setting of Northcutt et al. 2021). Notably, Cleanlab counts the samples with presumably incorrect noisy labels $\tilde{y}$ only, but does not provide any estimate about what LFs assigned these noisy labels (in contrast to ULF). More details are provided in Appendix B.

## 5 EXPERIMENTS

In order to demonstrate the effectiveness of our approach, we conducted several experiments of ULF and DeepULF on different datasets and tasks. The results are summarized in Tables 1 and 2; implementation details are provided in Appendix D[2].

**Datasets.** We evaluate feature-based ULF on four weakly supervised English datasets: (1) YouTube Spam Classification dataset (Alberto et al., 2015); (2) Spouse Relation Classification dataset (Corney et al., 2016); (3) Question Classification dataset from TREC-6 (Li & Roth, 2002); (4) SMS Spam Classification dataset (Almeida et al., 2011). For testing the DeepULF, we also added two topic classification datasets for African languages (5) Yorùbá and (6) Hausa (Hedderich et al., 2020). We use the same LFs and evaluation metrics as in previous works for a fair comparison. For all datasets except Spouse, the gold manual labels are available but used only for *Gold* baseline. The datasets and LFs descriptions as well as examples of LFs are provided in Appendix C.

**Baselines.** Feature-based ULF is compared against five baselines. (1) *Gold*: the classifier is trained with gold labels. As this is the only setting where manually obtained gold labels are used, it serves as an upper-bound baseline. (2) *Majority Vote*: the classifier is trained on the data and noisy labels acquired with simple majority voting and randomly broken ties. (3) *Snorkel-DP*: a classifier is trained using both generative and discriminative Snorkel steps (Ratner et al., 2020), (4) *CrossWeigh* (Wang et al., 2019c), (5a) *Cleanlab*: the original implementation of Northcutt et al. 2021 with Scikit-learn library, (5b) *Cleanlab-PyTorch*: our PyTorch-based reimplementation of the Cleanlab algorithm to provide a fair comparison with our methods implemented with the PyTorch library. DeepULF is compared towards seven baselines based on fine-tuning of a pre-trained language model. In all experiments, the pre-trained RoBERTa model (Liu et al., 2019) is used for the English datasets, and

---

[2]The code will be publicly released on acceptance.

the pre-trained multilingual BERT (Devlin et al., 2018) - for Yorùbá and Hausa following Hedderich et al. 2020. (1) *Gold*, (2) *Majority Vote*, and (3) *Snorkel-DP*: the same baselines as for the feature-based ULF, but with a pre-trained language model as the end model, (4) *FS + RoBERTa*: Flying Squid (Fu et al., 2020) is used as the label model and a pre-trained language model as the end model, (5) *Majority Vote + Cosine*, 6) *Snorkel-DP + Cosine*, and (7) *FS + Cosine*: the same as the baselines (2), (3) and (4), but applying the Cosine end model.

**Feature-based Experiments.** Table 1 reports the results of feature-based learning using a logistic regression classifier and TF-IDF data representations averaged over ten trials together with the standard error of the mean. Our WSCW and WSCL extensions of CrossWeigh and Cleanlab frameworks outperform the corresponding base methods in most cases and lead to consistent improvements compared to the Majority and Snorkel-DP baselines. These results support our initial hypothesis of LFs' importance in applying cross-validation techniques for weak supervision. At the same time, the ULF algorithm shows the best result overall on most datasets and even outperforms the model trained on YouTube data with gold manual annotations. $ULF_{sgn}$ outperforms the classifier trained on the data with labels chosen by majority voting without any additional denoising by 6.8% on average and Snorkel-DP by 9.2%, $ULF_{rndm}$ - by 4.3% and 6.7% respectively. Interestingly, the $ULF_{lfs}$ demonstrates a worse result compared to $ULF_{sign}$ and even $ULF_{rndm}$. That could be explained by the fact that more than one LF is matched in significant amounts of data samples in all datasets. As a result, these samples are included in several training and held-out folds (during the folds splitting in the cross-validation training according to the matched LFs) and therefore are reestimated multiple times, while ideally each data sample should be considered only once in each denoising round.

**Language model-based experiments.** DeepULF results are provided in Table 2. We did not perform a grid search for energy considerations and resource constraints but a random search with ten trials for each model type (DeepULF and baselines) with different hyperparameters. The search space and the best parameter settings are provided in Appendix D. We report the results for the model that was selected based on the performance on the development set.[3] For the baselines, we used the Wrench implementations (Zhang et al., 2021). We evaluated DeepULF in four variants which are determined by the combinations of two additional binary hyper-parameters: the first controls the model used for cross-validation (Alg. 1, line 7), the second controls the end model (Alg. 1, line 14); in both cases, the options are either simple fine-tuning or followed by additional Cosine training. Results of all cross-validation and end model combinations are provided in Appendix E, the best ones are included to Table 2. DeepULF outperforms the baselines on five out of six datasets and shows the second-best result for Yorùbá data. Note that both feature-based ULF and DeepULF do not involve manually annotated data or manual correction and cannot be directly compared to the results reported for settings that include manual annotations (Karamanolakis et al., 2021; Awasthi et al., 2020).

| | YouTube (Acc) | Spouse (F1) | TREC (Acc) | SMS (F1) | Yorùbá (F1) | Hausa (F1) | Avg |
|---|---|---|---|---|---|---|---|
| **Manually Supervised** | | | | | | | |
| Gold | 98.8 | - | 96.6 | 97.7 | 67.3 | 83.5 | 88.8 |
| **Weakly Supervised** | | | | | | | |
| Majority Vote | 93.2 | 21.3 | 68.6 | 93.0 | 48.1 | 43.9 | 61.4 |
| Snorkel-DP | 95.6 | 32.6 | 61.8 | 94.6 | **58.7** | 45.7 | 64.8 |
| FS (Fu et al., 2020) | 94.0 | 14.9 | 35.8 | 23.7 | 32.4 | 45.1 | 41.0 |
| **Cosine-based** (Yu et al., 2021) | | | | | | | |
| Majority Vote + Cosine | 96.4 | 33.3 | 65.8 | 93.6 | 52.6 | 45.4 | 64.5 |
| Snorkel-DP + Cosine | 96.0 | 28.1 | 73.8 | 96.1 | 55.0 | 46.5 | 65.9 |
| FS + Cosine | 95.6 | 24.9 | 38.6 | 90.1 | 33.3 | 41.5 | 54.0 |
| **DeepULF** (Ours) | **96.8** | **36.9** | **76.8** | **96.1** | 55.8 | **48.2** | **68.4** |

Table 2: DeepULF results. The best results are marked **bold**, the second best - underline.

---

[3]Trained models will be publicly released on acceptance.

## 6 CASE STUDY

For the case study, we chose a couple of samples with the wrong weak labels (i.e., the annotation done by labeling functions was incorrect) and check their labels after the ULF training. These examples are provided in Figure 4.

Sample 1 is clearly a SPAM comment. However, it is misclassified and assigned to the HAM class, because no LF corresponding to the SPAM class matched in it (this sample includes a spam keyword *"subscribe"*; however, as it is reduced to *"sub"*, the corresponding LF did not match), but a LF *short_comment* (labeling a sample as HAM if it is shorter than 5 words) did. Sample 2 is also a SPAM message not covered by any keyword-based LFs (as they are primarily concerned with cases where a user asks to subscribe to the channel, e.g. keywords *"subscribe"*, *"my"*, while here there is a request to *like* the comment). Nevertheless, again, a signal from the LF *short_comment* wrongly assigns this sample to the HAM class. Thus, the LF *short_comment* consistently mislabels samples that, although short, should still be classified as SPAM. ULF detects this LF and adjusts its assignment: it now corresponds to the HAM class with only roughly a 40% probability, and not 100% as before; with the remaining 60% it corresponds to the SPAM class. As a result, after label recalculation, both of these samples obtain a correct label SPAM.

In sample 3, a word *"subscribe"* (which is usually used in spam comments and, therefore, was leveraged to build a SPAM class LF *"keyword_subscribe"*) is mentioned, but in a context related to the video; This makes this message not-SPAM in contrast to the assigned label HAM in effect. In the same way, the keyword *"password"*, which is defined as a LF corresponding to the SPAM class in SMS dataset (in order to detect spam messages like *"Send me your id and password"*), is matched in a HAM sample 4 and annotates it with SPAM label. Again, ULF detects these noisy LFs and changes their assignment, which results in a change of the sample labels.

The samples where two LFs from different classes match can be assigned to either of the two classes with a 50% probability. In sample 5, the tie was broken randomly in favor of a wrong SPAM label. After the ULF reassignments, the LF *keyword_direct* was related to SPAM class with 70% instead of 100%. Although the assignment of LF *keyword_thanks* also changed slightly (90% to the HAM class and 10% to the SPAM class), overall the HAM class outweighed, and the sample label was changed to the correct one.

| | Sample | LFs matched | Noisy Label | Corrected Label | True Label |
|---|---|---|---|---|---|
| 1 | sub my channel for no reason -_- | short_comment | HAM | SPAM | SPAM |
| 2 | :):):) Like This Comment :):):) | short_comment | HAM | SPAM | SPAM |
| 3 | OMG I LOVE YOU KATY PARRY YOUR & SONGS ROCK!!!!!!!!!!!!!!!!!! THATS A TOTAL SUBSCRIBE | keyword_subscribe | SPAM | HAM | HAM |
| 4 | s...from the training manual it show there is no tech process:) its all about password reset and troubleshooting:) | keyword_password | SPAM | HAM | HAM |
| 5 | thanks for your message. i really appreciate your sacrifice. i'm not sure of the process of direct pay but will find out on my way back from the test tomorrow. i'm in class now. do have a wonderful day. | keyword_thanks keyword_direct | SPAM | HAM | HAM |

Figure 4: Examples of changed labels in YouTube and SMS datasets after denoising with ULF$_{sgn}$.

## 7 CONCLUSION

In our work, we explored the denoising of weakly supervised data using information from labeling functions that annotate the data. Our original intuition was that the noise specific to some LFs can be detected by training a model that does not use those LFs signals (i.e., the LFs are held-out) and then comparing the predictions of that model to the labels generated by the held-out LFs. A mismatch between such *out-of-sample* predictions and the *weak labels* could point to potentially incorrect labels produced by the noisy LF. This way, improving the weak labels can be done without involving any additional source of annotation or external knowledge but with the data itself.

This idea was realized in our new extensions of two $k$-fold cross-validation frameworks, initially proposed for denoising the manually annotated data (CrossWeigh and Cleanlab), as well as in our new method **ULF** for unsupervised labeling functions denoising. The extensive experiments with feature-based ULF and DeepULF demonstrate the effectiveness of our approach and confirm the significant role of labeling functions in weakly supervised data denoising.

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

## A  WEAKLY SUPERVISED CROSSWEIGH

The original CrossWeigh framework (CW, Wang et al. 2019c) was proposed for tracing inconsistent labels in the crowdsourced annotations for the NER task. After randomly splitting the data into $k$ folds and building $k$ training and hold-out sets, CrossWeigh additionally filters the training samples that include the entities matched in hold-out folds samples. The intuition behind this approach is: if an entity is constantly mislabeled, the model would be misguided; but the model trained without it would be rid of this confusion.

We consider this approach quite promising for detecting unreliable LFs in weakly supervised data in a similar way. If a potentially erroneous LF systematically annotates the samples wrongly, a reliable model trained data without it will not make this mistake in its prediction, and, thus, the error will be traced and reduced. Our new **Weakly Supervised CrossWeigh** method (WSCW) allows splitting the data not entirely randomly but considering the LFs so that all LFs matched in a test fold are eliminated from the training folds.

More formally, we, firstly, randomly split labeling functions $L$ into $k$ folds $\{f_1, ..., f_k\}$. Then, we iteratively take LFs from each fold $f_i$ as test LFs and the others as training LFs. So, all samples where no LFs from hold-out fold match become training samples, while the rest are used for testing.

$$X_{train_i} = \{x_j | L_{x_j} \cap f_i = \emptyset\}$$
$$X_{out_i} = X \setminus X_{train_i}$$

After that we train the $k$ separate models on $X_{train_i}$ and evaluate them on $X_{out_i}$. In the same way as in the original CrossWeigh algorithm, the labels predicted by the trained model for the samples in the hold-out set $\hat{y}$ are compared to the initial noisy labels $y$. All samples $X_j$ where $\hat{y}_j \neq y_j$ are claimed to be potentially mislabeled; their influence is reduced in further training. The whole procedure of error detection is performed $t$ times with different partitions to refine the results. The sample weights $w_{x_N}$ are then calculated as $w_{x_j} = \epsilon^{c_j}$, where $c_j$ is the number of times a sample $x_j$ was classified as mislabeled, $0 \leq c_j \leq t$, and $\epsilon$ is a weight reducing coefficient.

## B  WEAKLY SUPERVISED CLEANLAB

The second method we introduce is **Weakly Supervised Cleanlab** (WSCL) - an adaptation of Cleanlab framework Northcutt et al. (2021) for weak supervision. In the same way as in WSCW, not data samples, but the labeling functions $L$ are split into $k$ folds $\{f_1, ..., f_k\}$ and used for building the $X_{train_i}$ and $X_{out_i}$ sets, $1 < i < k$, for training $k$ models.

In contrast to WSCW, for each sample $x_i$ the label is not directly predicted on the $X_{out_i}$, but the probability vector of class distribution $\hat{p}(y = j; x_i, \theta), j \in K$ is considered. The exact labels $\hat{y}$ are calculated later on with respect to the class expected self-confidence value $t_j$ (see Northcutt et al. 2021):

$$t_j := \frac{\sum_{X_j} \hat{p}(y = j; x_i, \theta)}{|X_j|}, \tag{8}$$

where $X_j = \{x_i \in X_{y=j}\}, 1 < j < c$

A sample $x_i$ is considered to confidently belong to class $j \in K$ if the probability of class $j$ is greater than expected self-confidence for this class $t_j$ or the maximal one in the case of several classes is probable:

$$\hat{y}_i = \underset{\substack{j \in [K]: \\ \hat{p}(y=j; x_i, \theta) \geq t_j}}{\arg \max} \hat{p}(y = j; x_i, \theta) \tag{9}$$

The samples with no probability that would exceed the thresholds have no decisive label and do not participate in further denoising.

After that, a class-to-class confident joint matrix $C_{y,\hat{y}}$ is calculated, where:

$$C_{y,\hat{y}}[j][k] = |\{x_i \in X | y_i = j, \hat{y}_i = j\}|$$

Notably, $C_{y,\hat{y}}$ contains only the information about correspondence between noisy and out-of-sample predicted labels (the same way as in Northcutt et al. 2021). So, it gives the idea about the number of samples with presumably erroneous noisy labels $y$ but does not give us any insights about the erroneous labeling functions that assigned this noisy label to this sample (in contrast to the ULF approach we present in Section 3).

The confident matrix $C_{y,\hat{y}}$ is then calibrated and normalized in order to obtain an estimate matrix of the joint distribution between noisy and out-of-sample predicted labels $\hat{Q}_{y,\hat{y}}$, which determines the number of samples to be pruned. We perform the pruning by noise rate following the Cleanlab default setting: $n \cdot Q_{y_i,\hat{y}_j}, i \neq j$ samples with $max(\hat{p}(y = j) - \hat{p}(y = i))$ are eliminated in further training.

## C  DATASETS

**YouTube**   (Alberto et al., 2015) A spam detection dataset was collected from the YouTube video comments. The samples that are not relevant to the video (e.g. advertisement of user's channel or ask for subscription) are classified as SPAM, while others belong to the HAM class. We use the same labeling functions as in (Ratner et al., 2020); they were created using keywords, regular expressions, and heuristics. For example, there is a labeling function `keyword_my` which corresponds to class SPAM, meaning that if a sample contains the word "my", it will be assigned to the SPAM class. Examples of other labeling functions are `KEYWORD_SUBSCRIBE`, `KEYWORD_PLEASE`, `KEYWORD_SONG` (all are the keyword-based), `SHORT_COMMENT` (i.e., if a YouTube comment is short, it is probably not spam; thus, samples less than 5 words long would be classed as HAM) and so on.

**Spouse**   (Corney et al., 2016) A relation extraction dataset based on the Signal Media One-Million News Articles Dataset, which main task is to define whether there is a spouse relation in a sample. We use the Snorkel annotation (Snorkel); the labeling functions were created based on keywords (e.g. `spouse, husband`), spouse relationships extracted from DBPedia (Lehmann et al., 2014), and language patterns (e.g. check whether the person mentioned in a sample have the same last name).

**TREC**   (Li & Roth, 2002) A question classification dataset that maps each data sample to one of 6 classes. The labeling functions were generated based on keywords (Awasthi et al., 2020), e.g. `which, what, located, situated` keywords relate a sample to the class `LOCATION`.

**SMS** (Almeida et al., 2011) A spam detection dataset comprised of text messages. The annotation (Awasthi et al., 2020) includes keyword-based and regular expression-based labeling functions. For example, a regex-based labeling function:

```
( |^)(won|won)[^\w]* ([^\s]+ )*(claim,|claim)[^\w]*( |$)
```

corresponds to class SPAM (e.g. as in a sample *449050000301 You have won a ??2,000 price! To claim, call 09050000301.*).

**Yorùbá and Hausa** (Hedderich et al., 2020) Topic classification datasets of the second (Hausa) and the third (Yorùbá) most spoken languages in Africa comprised of news headlines. The weak keyword rules are provided by the authors (Hedderich et al., 2020).

|  | YouTube | Spouse | TREC | SMS | Yorùbá | Hausa |
|---|---|---|---|---|---|---|
| Train Data | 1586 | 22254 | 4965 | 4502 | 1340 | 2045 |
| Valid Data | 150 | 2711 | 500 | 500 | 189 | 290 |
| Test Data | 250 | 2701 | 500 | 500 | 379 | 582 |
| #Classes ($K$) | 2 | 2 | 6 | 2 | 7 | 5 |
| #LFs ($L$) | 10 | 9 | 68 | 73 | 19897 | 18624 |
| #Unlabeled | 195 | 16520 | 242 | 2719 | 0 | 0 |
| Avg LF Hits | 1.6 | 33.7 | 1.7 | 0.5 | 3.0 | 2.9 |
| LF Accuracy | $81\% \pm 2.0$ | $53\% \pm 0.6$ | $50\% \pm 2.6$ | $60\% \pm 1.6$ | $55\% \pm 1.5$ | $54\% \pm 0.3$ |
| LF Coverage | 87% | 25% | 85% | 40% | 100% | 100% |

Table 3: Statistics of all the datasets. The LF accuracy metrics are calculated with majority vote without any model training reported across 5 runs with standard deviation in order to reduce the instability caused by randomly broken ties.

## D  IMPLEMENTATION DETAILS

**Feature-based ULF.** The feature-based ULF is realized with logistic regression model; the training data are encoded with TF-IDF vectors. For training, we set the number of epochs to 20 and applied the early stopping with patience = 5. Each experiment was run 10 times with different initializations. Development set is used for early stopping as well as to estimate the number of iterations $I$: initially, it is set to $I = 20$, but if training labels do not change after three iterations, the algorithm stops, and the last saved model is used for final testing. The actual number of iterations in training of the best performing ULF model ULF$_{sng}$, alongside other hyperparameter values found using grid-search, can be found in Table 10; the search space is provided in Table 4.

| Hyperparameter | Values |
|---|---|
| Multiplying coefficient $p$ | 0.1, 0.2, 0.3, 0.4, 0.5, 0.7, 0.9 |
| Learning rate $lr$ | 1e-1, 1e-2, 1e-3, 1e-4 |
| Number of folds $k$ | 3, 5, 8, 10, 15, 20 (with respect to the overall number of LFs) |
| Non-labeled data rate $\lambda$ | 0, 0.5, 1, 2, 3 |

Table 4: Feature-based ULF: hyperparameter values tried in grid search.

|  | YouTube | Spouse | TREC | SMS |
|---|---|---|---|---|
| Multiplying coefficient $p$ | 0.5 | 0.2 | 0.3 | 0.1 |
| Learning rate $lr$ | 1e-2 | 1e-2 | 1e-1 | 1e-1 |
| Number of folds $k$ | 8 | 3 | 3 | 10 |
| Number of iterations I | 5 | 1 | 1 | 2 |
| Non-labeled data rate $\lambda$ | 0 | 3 | 1 | 0.5 |

Table 5: ULF$_{sng}$ selected hyperparameters.

**DeepULF.** The pre-trained language models were downloaded from HuggingFace[4]. To encode the features, we used a multilingual pre-trained BERT model[5] for Yoruba and Hausa datasets and pre-trained RoBERTa model[6] for others and followed the Wrench encoding method (Zhang et al., 2021). The gradient-based optimization was performed with AdamW Optimizer and linear learning rate scheduler, and the early stopping was applied based on the evaluation metric values on the validation set. For the baselines and Cosine training, we took the parameters listed by zhang2021wrench. Additionally, we include the label prediction parameter: if it equals "soft", the probabilistic labels are used for training; otherwise ("hard") a label is the one-hot encoding of the most probable class (the ties are broken randomly). ULF-specific parameter search space was defined heuristically. All parameters are provided in Table 6.

In order to reduce computational load we used the random parameter search instead of grid search. Specifically, we tried 10 random parameter combinations from search space and selected the one which performed the best on the development set.

| Hyperparameter | Values |
|---|---|
| Multiplying coefficient $p$ | 0.1, 0.3, 0.5, 0.7, 0.9 |
| Learning rate $lr$ | 1e-4, 1e-5, 1e-6 |
| Number of folds $k$ | 3, 5, 7 (with respect to the overall number of LFs) |
| Number of iterations $I$ | 1, 2, 3, 4 |
| Confident regularization weight $\lambda$ | 0.01, 0.1 |
| The confident threshold $\xi$ | 0.2, 0.4, 0.6, 0.8 |
| Label prediction | soft, hard |

Table 6: DeepULF: hyperparameter values tried in random search.

|  | YouTube | Spouse | TREC | SMS | Yorùbá | Hausa |
|---|---|---|---|---|---|---|
| Multiplying coefficient $p$ | 0.5 | 0.1 | 0.1 | 0.3 | 0.1 | 0.1 |
| Learning rate $lr$ | 1e-4 | 1e-06 | 1e-05 | 1e-05 | 1e-06 | 1e-06 |
| Number of folds $k$ | 2 | 3 | 3 | 3 | 2 | 2 |
| Number of iterations I | 1 | 2 | 1 | 1 | 2 | 2 |
| Confident regularization weight $\lambda$ | 0.1 | 0.01 | 0.1 | 0.1 | 0.1 | 0.05 |
| Confident threshold $\xi$ | 0.8 | 0.2 | 0.6 | 0.8 | 0.2 | 0.8 |
| Label prediction | soft | soft | soft | soft | soft | soft |

Table 7: DeepULF$_{FT\_FT}$ selected hyperparameters.

---

[4]https://huggingface.co/models

[5]https://huggingface.co/bert-base-multilingual-cased

[6]https://huggingface.co/roberta-base

| | YouTube | Spouse | TREC | SMS | Yorùbá | Hausa |
|---|---|---|---|---|---|---|
| Multiplying coefficient $p$ | 0.5 | 0.1 | 0.3 | 0.3 | 0.1 | 0.1 |
| Learning rate $lr$ | 1e-4 | 1e-06 | 1e-06 | 1e-05 | 1e-05 | 1e-05 |
| Number of folds $k$ | 2 | 3 | 5 | 3 | 2 | 2 |
| Number of iterations I | 1 | 3 | 1 | 1 | 2 | 1 |
| Confident regularization weight $\lambda$ | 0.1 | 0.01 | 0.01 | 0.1 | 0.1 | 0.05 |
| Confident threshold $\xi$ | 0.8 | 0.2 | 0.8 | 0.8 | 0.2 | 0.4 |
| Label prediction | soft | hard | soft | soft | soft | soft |

Table 8: DeepULF$_{FT\_COS}$ selected hyperparameters.

| | YouTube | Spouse | TREC | SMS | Yorùbá | Hausa |
|---|---|---|---|---|---|---|
| Multiplying coefficient $p$ | 0.1 | 0.1 | 0.1 | 0.3 | 0.1 | 0.1 |
| Learning rate $lr$ | 1e-06 | 1e-06 | 1e-05 | 1e-06 | 1e-06 | 1e-05 |
| Number of folds $k$ | 3 | 3 | 3 | 5 | 7 | 2 |
| Number of iterations I | 2 | 1 | 1 | 2 | 4 | 1 |
| Confident regularization weight $\lambda$ | 0.1 | 0.01 | 0.1 | 0.01 | 0.05 | 0.05 |
| Confident threshold $\xi$ | 0.2 | 0.2 | 0.6 | 0.4 | 0.4 | 0.4 |
| Label prediction | soft | hard | soft | soft | soft | soft |

Table 9: DeepULF$_{COS\_FT}$ selected hyperparameters.

| | YouTube | Spouse | TREC | SMS | Yorùbá | Hausa |
|---|---|---|---|---|---|---|
| Multiplying coefficient $p$ | 0.7 | 0.1 | 0.3 | 0.1 | 0.1 | 0.1 |
| Learning rate $lr$ | 1e-05 | 1e-06 | 1e-06 | 1e-05 | 1e-06 | 1e-05 |
| Number of folds $k$ | 5 | 3 | 5 | 7 | 7 | 2 |
| Number of iterations I | 4 | 2 | 2 | 1 | 1 | 2 |
| Confident regularization weight $\lambda$ | 0.01 | 0.01 | 0.01 | 0.05 | 0.05 | 0.05 |
| Confident threshold $\xi$ | 0.2 | 0.2 | 0.8 | 0.4 | 0.4 | 0.4 |
| Label prediction | soft | hard | soft | hard | soft | hard |

Table 10: DeepULF$_{COS\_COS}$ selected hyperparameters.

Both feature-based ULF and DeepULF were implemented with Python and PyTorch (Paszke et al., 2019) in the setting of the weak supervision framework *Knodle* (Sedova et al., 2021). By providing an access to all WS components Knodle allowed us to implement and benchmark all algorithms described above. ULF experiments were performed on a machine with a CPU frequency of 2.2GHz with 40 cores; DeepULF experiments were run on asingle Tesla V100 GPU on Nvidia DGX-1. The full setup took on average 20 hours for each dataset for ULF and 96 hours for DeepULF.

# E  DEEPULF RESULTS

| | YouTube (Acc) | Spouse (F1) | TREC (Acc) | SMS (F1) | Yorùbá (F1) | Hausa (F1) |
|---|---|---|---|---|---|---|
| DeepULF$_{FT\_FT}$ | 96.8 | 22.0 | 68.2 | 96.1 | 54.6 | 43.0 |
| DeepULF$_{FT\_Cos}$ | 94.4 | 36.9 | 76.6 | 96.1 | 54.2 | 48.2 |
| DeepULF$_{Cos\_FT}$ | 95.2 | 21.3 | 68.6 | 96.1 | 55.8 | 43.6 |
| DeepULF$_{Cos\_Cos}$ | 94.8 | 33.0 | 76.8 | 96.1 | 54.2 | 44.5 |

Table 11: DeepULF results. All possible cross-validation and end model combinations.

