# OpenReview forum: "ULF: UNSUPERVISED LABELING FUNCTION CORRECTION USING CROSS-VALIDATION FOR WEAK SUPERVISION"
_ICLR.cc/2023/Conference — Submitted to ICLR 2023_

### Official Review · Reviewer_Cajh · 2022-10-25

**Confidence:** 3
**Correctness:** 3
**Technical Novelty And Significance:** 3
**Empirical Novelty And Significance:** 3
**Recommendation:** 6

**Clarity, Quality, Novelty And Reproducibility:**

* Clarity: the paper is quite difficult to read as the cross-validation-based confidence estimation description is quite dense. I found fig 3 very helpful for understanding the whole ULF. The experiments results and case study are quite lengthy and not really helpful. They should be shorter to make room for analyses.

* Quality: the paper is well motivated, clear and meaning contribution, and well supported by experiments.

* Originality: the paper is novel.

**Strength And Weaknesses:**

ULF is an interesting and effective way to use confidence estimation in weakly supervised learning setting.

* It is well motivated, for why labelling labels can be noisy and how correcting the noise can be helpful. Cross-validation-based confidence estimation is well argued as a promising way to overcome the problem. However, using this confidence estimation is not straightforward (as pointed out in the experiments -- comparison against wscrossweigh and wscleanlab).

* The idea of incrementally updating the mapping between labelling functions and classes seems to have a nice connection with the noisy-channel-based approaches (e.g. Goldberger & Ben-Reuven 2017) where confusion matrices play a key role in estimating the labelling noise.

The paper however lacks a theory to support the idea. Incrementally updating the matrix T* and noise labels Y* sounds reasonable but can be dangerous as we don't know to where they will converge (or will they converge at all). In fact, there's even no objective for ULF to work on. (On the contrary, the noisy-channel matrix is part of the loss to be minimized.) Would it be possible to view T* and Y* as latent variables and put them in a theoretical framework (as in Snorkel or noisy-channel)?

When comparing ULF to data programming Snorkel, it is unclear whether ULF is able to deal with dependency between labelling functions.

The use of development sets in experiments for early stopping and hyper param tuning is typical. However, if these sets are clean (annotated labels are not noisy), then they don't seem to be used efficiently. For instance, why not using them to estimate the confidence directly? Or, at least they maybe more helpful than only to select p. Also, having clean dev. sets can be tricky as one can train a model directly on these sets in a supervised learning fashion.

As estimating T* and Y* is the core of ULF, the paper should have some analyses on whether T* and Y* are estimated more correctly after each iteration.



**Summary Of The Paper:**

The paper proposes a method called ULF to correct the mapping between labelling functions and classes in weakly supervised learning. In this setting, initially a labelling function maps instances to some specific classes. As this mapping can be noisy, ULF iteratively updates the (soft) mapping by estimating how correct the mapping is. This estimation is based on cross-validation-based confidence learning proposed by Northcutt et al (2022). The paper demonstrates that ULF outperforms several weakly supervised baselines (including Snorkel-DP) on 4 text classification datasets.

**Summary Of The Review:**

In general I like the paper, both the ideas and the experimental results. It should be a welcomed contribution to the community whose work is on learning from noisy data.

Although I recommend to accept the paper, there are several points that the paper can be improved:
* the writing can be clearer (especially the description for cross-validation-based confidence estimation), some parts can be shorten for analyses.

* the paper will be stronger if ULF can be framed in a theoretical framework.

* more analyses are needed to understand the quality of T* and Y*.

---

> ### Author Response · Authors · 2022-11-18
> **Thank you for your feedback!**
>
> Thank you for your positive feedback! Your questions are answered below.
>
> 1) We view labeling functions improvement through the lens of data quality, and we built ULF on ideas from data quality improvement (but, importantly, ULF makes corrections to the labeling function-to-class assignment, rather than to the labels, since the labeling functions are the source of error in weak supervision). We expect that many of the theoretical results from data quality improvement (CleanLab [1]) could be carried over to our setup although we didn’t attempt a formal analysis.
>
> 2) Labeling function correlation is reflected in ULF via the feature space (and the trained classifiers), and not via estimation of dependencies from direct cooccurrences/correlations (as in Snorkel). Therefore, ULF provides a “smoother” version of LF “dependencies” than Snorkel, because LFs do not need to co-occur in the same samples of the training data to be detected as similar/agreeing.
>
> 3) ULF does not use manual annotations for denoising the weak data, and the dev data (the only gold data exploited in our experiments) is never used for improving the noisy data or for training the classifier. In general, the data is used as slit as in previous works.
>
> 4) The reestimated T matrices (compared to the initial ones) will be added to Appendix in order to demonstrate the exact changes made by ULF.
>
> [1] Curtis G. Northcutt, Lu Jiang, and Isaac L. Chuang. Confident learning: Estimating uncertainty in dataset labels. Journal of Artificial Intelligence Research (JAIR), 2021

---

### Official Review · Reviewer_SFmE · 2022-10-25

**Confidence:** 4
**Correctness:** 2
**Technical Novelty And Significance:** 2
**Empirical Novelty And Significance:** 2
**Recommendation:** 5

**Clarity, Quality, Novelty And Reproducibility:**

- I have some concerns on the reported numbers (replicating WRENCH and reported above)
- I don't think the paper, overall, is particularly novel. There are virtues to a simple method for improving performance, say an deep empirical dive into k-fold estimation when designing labeling functions, but this manuscript doesn't provide enough details to be very illuminating.
- Code will be released, but is not currently available

**Strength And Weaknesses:**

Strengths
-  Viewing LF improvement through the lens of data quality is interesting and methods to fuse techniques that can modify and repair LFs is a valuable research focus
- The authors use the WRENCH benchmark, which addresses an endemic weakness in weak supervision papers where the labeling functions are not consistent across methods, confounding performance insights.

Weaknesses
- The stated goal of the paper is to "primary goal is to improve the LFs to classes allocation in order to correct the systematically biased label assignments". This targets fundamentally misspecified labeling functions, since a core assumption of vanilla DP is that LFs perform better than random chance. These can happen in practice, but I'm not convinced flipping the label (essentially) is always beneficial especially in rare/infrequent classes settings.
- The paper suffers from clarity issues. For example, the feature-based and deep ULF methods need more methods elaboration and details
- From the Table 2 results, which use the WRENCH benchmark implementations (https://arxiv.org/pdf/2109.11377.pdf) the reported performance numbers don't seem to match what was reported in the original WRENCH paper. See their Table 12 the best reported best results for RoBERTa + Cosine given various label model classes. I've repeated the key parts below. This paper's implementation is underperforming and DeepULF does not outperform a tuned model. My concern is that the perform benefits might simply wash out when using a tuned end model.
| Dataset/Metric | WRENCH/Label Model | This Paper | DeepULF |
|----------------|--------------------|------------|---------|
| YouTube/Acc    | 97.60 / MV         | 96.4       | 96.8    |
| Spouse/F1      | 40.50 / MV         | 33.3       | 36.9    |
| TREC/ACC       | 77.96 / MV         | 65.8       | 76.8    |
| SMS/F1         | 96.67 / MV         | 93.6       | 96.1    |
- There are stronger baseline label models than Flying Squid and the default Data Programming (Snorkel-DP) model See Mazzetto et al 2021. "Semi-Supervised Aggregation of Dependent Weak Supervision Sources With Performance Guarantees"
- Ideally this type of paper would be more systematic in exploring properties of the task themselves, including low frequency class problems, higher cardinalities. It's hard to draw any strong insights from the experiments as presented.

**Summary Of The Paper:**

This paper presents ULF (Unsupervised Labeling Function) which uses k-fold cross-validation to correct errors in labeling functions (LFs) by using cross-validation to identify confident sample estimates and update LF class label assignments. They explore 2 variants of ULF: a count-based and language model / DeepULF based method. The evaluate on up to 6 benchmark tasks and explore a wide range of baseline methods, including two other cross-validation-based methods and a host of label model configurations used to train RoBERTa + Cosine (an existing self training method for incorporating unlabeled training data). The manuscript reports that ULF in it's variations outperforms existing methods.

**Summary Of The Review:**

Unfortunately given the clarity in presentation and the inconsistency with reported results (which would erase ULF's purported benefits), I don't think this paper is ready for acceptance.

---

> ### Author Response · Authors · 2022-11-18
> **Thank you for your feedback and suggestions!**
>
> Thank you for your feedback. We answered your questions as below.
>
> 1) Note that we do not do a hard flip on the labels but rather adjust/loosen their assignment to one particular class. Our motivation does not contradict the assumption of LFs better performance compared to the random chance. However, using the hard assignment as it is done in naive weak supervision (i.e., a labeling function either corresponds to this class or not; in other words, it is either 100% or 0% probability that a labeling function corresponds to some class) can potentially cause problems such as the ones that are presented in Figure 1. ULF changes such initial hard assignment to a more smooth fine-adjusted one. As the result, different LFs participate according to their strength in association with the classes in the later label voting.  Our experiments showed that this improved performance overall, but of course, ULF cannot guarantee improvement for every LF or example.
>
> 2) WRENCH numbers were obtained in a large-scale effort of exhaustively searching the entire parameter range by a brute-force grid search. However, unfortunately neither resulting  models nor hyperparameters were  published by the WRENCH authors (they were also not able to provide us with either of that on our request as well). Running an extensive grid search (just for the baselines) would be computationally prohibitive, so for a fair comparison, we opted to do a random search (for baselines and our methods) in the range reported in WRENCH, running the WRENCH code. We will publish the models that produced the results for the baselines and our methods upon acceptance/de-anonymization (as indicated in Footnote 3).
>
> 3) Mazzetto et al (2021) do not compare their method on the standard weakly supervised datasets (e.g. the ones that are used in WRENCH). Instead, we compare several established datasets and methods that performed well in the more recent and more extensive WRENCH benchmark.
>
> 4) The code had in fact been uploaded to openreview at submission time - please check it again. We did not open-source the code yet to preserve the anonymity of our submission, but we will do it after acceptance, as indicated in Footnote 2.

---

### Official Review · Reviewer_ECG9 · 2022-10-26

**Confidence:** 4
**Correctness:** 3
**Technical Novelty And Significance:** 3
**Empirical Novelty And Significance:** 3
**Recommendation:** 5

**Clarity, Quality, Novelty And Reproducibility:**

Clarity, Quality: The paper is well written and easy to follow.
Novelty: The paper has its merit but still some claims need to be further validated.
Reproducibility: The authors provide psuedo code and implementation description. The code is included in the supplementary.

**Strength And Weaknesses:**

Strength:
1. The authors work on a practical and important problem.
2. The authors introduce their method with a clear description and motivation
3. The proposed method mainly aims to refines the allocation of LFs to classes, which is a reasonable and less studied design.

Weakness:
1. The [1] shares a similar motivation, which also aims to learn the weights of LFs to data.
2. The reported performance on TREC from [2] is 82.59, which is higher than that of the proposed method and its implementation in Table 2.  It is suggested to add more experiments on the some datasets from  [2] to carefully validate the progress.
3. The motivation is clear but corresponding empirical analysis are missing. What are the refine results of the allocation of LFs to classes?

[1] Giannis Karamanolakis, Subhabrata Mukherjee, Guoqing Zheng, and Ahmed Hassan Awadallah. Self-training with weak supervision.  NAACL 2021.
[2] Yue Yu, Simiao Zuo, Haoming Jiang, Wendi Ren, Tuo Zhao, Chao Zhang. Fine-Tuning Pre-trained Language Model with Weak Supervision: A Contrastive-Regularized Self-Training Approach NAACL 2021

**Summary Of The Paper:**

In this paper, the authors investigate noise reduction techniques for weak supervision based on the principle of k-fold cross-validation.  More specifically, the authors propose a new algorithm ULF for denoising weakly annotated data which uses models trained on all but some LFs to detect and correct biases specific to the held-out LFs. The experimental results help validate the effectiveness of  ULF in weakly supervised learning setting.

**Summary Of The Review:**

The authors propose a method ULF, which could denoise weakly annotated data by refining the allocation of LFs to classes. The motivation of the proposed method is clear and the proposed design is reasonable. The experiments are conducted on several datasets to validate the effectiveness. Some of concerns are included in the Strength And Weaknesses section.

---

> ### Author Response · Authors · 2022-11-18
> **Thank you for your feedback and suggestions!**
>
> Thank you for your feedback. We answered your questions as below.
>
> 1) Indeed, paper [1] works on a similar problem. We cite it in related works and highlight its main difference from our algorithm, which is: the authors of [1] use an additional set of manually labeled data, while ULF doesn’t use any manual supervision at all. In other words, the ASTRA framework presented in [1] builds a semi-supervised system that leverages (among others) weakly annotated data, while ULF aims at training a classifier *exclusively* on the automatically labeled denoised weak data.
>
> 2) In paper [2] the authors apply grid search for retrieving the best hyperparameter combination, while we used random search/hyperparameter sampling for energy and resource considerations (as we mention in Section 5). In order to provide a fair comparison, a random search was executed for all baselines and for our method as well. This might be the cause of the lower baseline numbers in our paper in comparison with [2]. Please also note that we use the WRENCH implementation for all the baselines, including the ones that exploit the method from [2].
>
> 3) We provide some examples of refined labeling functions and weak labels in Section 6 (Case study). We also discuss the potential root cause of the initial wrong assignment and the reason why ULF decided to change class assignment/label. The re-estimated T matrices (compared to the initial ones) will be added to Appendix in order to demonstrate the exact changes made by ULF.
>
> [1] Giannis Karamanolakis, Subhabrata Mukherjee, Guoqing Zheng, and Ahmed Hassan Awadallah. Self-training with weak supervision. NAACL 2021.
> [2] Yue Yu, Simiao Zuo, Haoming Jiang, Wendi Ren, Tuo Zhao, Chao Zhang. Fine-Tuning Pre-trained Language Model with Weak Supervision: A Contrastive-Regularized Self-Training Approach NAACL 2021

---

### Official Review · Reviewer_jWdL · 2022-10-28

**Confidence:** 3
**Clarity, Quality, Novelty And Reproducibility:** The paper is well written, clear and …
**Correctness:** 3
**Technical Novelty And Significance:** 2
**Empirical Novelty And Significance:** 2
**Recommendation:** 5

**Strength And Weaknesses:**

Strengths:
- The paper has strong performance improvements compared with the baseline and competitor methods, as listed in table 1 and table 2
- The method is relatively simple to implement for denoising automatic labels and could have large scale impact if well generalizable

Weaknesses:
- The method is based on an intuition that a “mismatch between predictions of a model and labels can indicate candidates of noise specific to all held out labeling functions.” This intuition needs stronger support and explanation beyond a few examples mentioned in Figure 1.
- It is unclear on whether this method can be applied beyond the language modality, and if so, how would that be implemented


**Summary Of The Paper:**

This paper provides a method for noise reduction in cases of automatic annotation using pre-defined labeling functions. They propose to do the noise reduction using the principle of k-fold cross validation, and through the iterative process are able to automatically denoise the entire dataset in an unsupervised fashion. Their major contribution is in more accurate labels and better quality of the trained classifier from the labels, and in providing a feature-based method (for feature based learning without a hidden layer) and deep learning method (for fine-tuning pre-trained language models).

**Summary Of The Review:**

The work is quite interesting and the idea is simple, but I would like to see more justification and examples for the intuition and why it makes sense.

---

> ### Author Response · Authors · 2022-11-18
> **Thank you for your feedback and suggestions!**
>
> Thank you for your feedback. Your questions are answered below.
>
> 1) Please note that the examples in Figure 1 rather demonstrate the potentially problematic weak annotations generally than explain the intuition behind ULF specifically. However, the examples and analysis of the mistakes caused by noise specific to the held-out labeling functions (which ULF aims at correcting) are provided in the ablation study (Figure 4). The wrong labels for these samples were corrected by ULF; in the corresponding section, we also explain how ULF traced these errors and corrected them (yet, of course, without a guarantee of 100% successful correction).
>
> 2) In principle, there are no limitations to the ULF method on modality. However, weak supervision and data programming are particularly relevant for natural language processing since it is natural to express intuition and find prototypical patterns (which can be used as labeling functions)  for language. That was our motivation to choose the language datasets and make our experiments in the language domain. Still, if you can express primary knowledge with labeling functions in any other domain, you can use ULF there as well.

---

### Decision · Program_Chairs · 2023-01-20

**Decision:**

Reject

**Justification For Why Not Higher Score:**

The paper suffers from the lack of clarity, theoretical justification as well as analysis of experimental results, which are also raised as issues by the reviewers. The authors responded to some of the reviewers' comments. However, the paper still needs significant work, particularly from the perspective of clarity and experimental analysis.

**Justification For Why Not Lower Score:**

N/A

**Metareview: Summary, Strengths And Weaknesses:**

The paper presents ULF (Unsupervised Labeling Function) correction, which uses k-fold cross-validation to reduce noise in weakly annotated data by a predefined set of labeling functions (LFs). Two realizations of the ULF were introduced in the paper. The proposed methods were benchmarked on tasks against similar methods from the literature showing the performance improvements due to ULF. Although the k-fold cross-validation method is not new for reducing label noise in weak supervision, its use in this specific setup is.  Overall, the ULF and its two realizations are novel, the paper suffers from the lack of clarity, theoretical justification as well as analysis of experimental results, which are also raised as issues by the reviewers. The authors responded to some of the reviewers' comments. However, the paper still needs significant work, particularly from the perspective of clarity and experimental analysis.